# Chloroquine Potentiates the Anticancer Effect of Pterostilbene on Pancreatic Cancer by Inhibiting Autophagy and Downregulating the RAGE/STAT3 Pathway

**DOI:** 10.3390/molecules26216741

**Published:** 2021-11-08

**Authors:** Rong-Jane Chen, Yi-Jhen Lyu, Yu-Ying Chen, Yen-Chien Lee, Min-Hsiung Pan, Yuan-Soon Ho, Ying-Jan Wang

**Affiliations:** 1Department of Food Safety/Hygiene and Risk Management, College of Medicine, National Cheng Kung University, Tainan 70403, Taiwan; janekhc@gmail.com; 2Department of Environmental and Occupational Health, College of Medicine, National Cheng Kung University, Tainan 70403, Taiwan; won491729@hotmail.com (Y.-J.L.); 101312123@gms.tcu.edu.tw (Y.-Y.C.); 3Department of Medical Oncology, Tainan Hospital, Ministry of Health and Welfare, Executive Yuan, Tainan 70043, Taiwan; yc_lee@post.harvard.edu; 4Department of Internal Medicine, College of Medicine, National Cheng Kung University Hospital, Tainan 70403, Taiwan; 5Institute of Food Science and Technology, National Taiwan University, Taipei 10617, Taiwan; mhpan@ntu.edu.tw; 6TMU Research Center of Cancer Translational Medicine, Taipei Medical University, Taipei 11031, Taiwan; 7Cancer Research Center, Taipei Medical University Hospital, Taipei 11031, Taiwan; 8School of Medical Laboratory Science and Biotechnology, College of Medical Science and Technology, Taipei Medical University, Taipei 11031, Taiwan; 9Department of Medical Research, China Medical University Hospital, China Medical University, Taichung 40402, Taiwan; 10Master Degree Program in Toxicology, College of Pharmacy, Kaohsiung Medical University, Kaohsiung 80708, Taiwan

**Keywords:** pterostilbene, pancreatic ductal adenocarcinoma, chloroquine, RAGE/STAT3, autophagy, apoptosis

## Abstract

The treatment of pancreatic ductal adenocarcinoma (PDAC) remains a huge challenge, because pro-survival signaling pathways—such as the receptor for advanced glycation end products (RAGE)/signal transducer and activator of transcription 3 (STAT3) pathway—are overexpressed in PDAC cells. Moreover, PDAC cells are highly resistant to chemotherapeutic agents because of autophagy induction. Therefore, autophagy and its modulated signaling pathways are attractive targets for developing novel therapeutic strategies for PDAC. Pterostilbene is a stilbenoid chemically related to resveratrol, and has potential for the treatment of cancers. Accordingly, we investigated whether the autophagy inhibitor chloroquine could potentiate the anticancer effect of pterostilbene in the PDAC cell lines MIA PaCa-2 and BxPC-3, as well as in an orthotopic animal model. The results indicated that pterostilbene combined with chloroquine significantly inhibited autophagy, decreased cell viability, and sensitized the cells to pterostilbene-induced apoptosis via downregulation of the RAGE/STAT3 and protein kinase B (AKT)/mammalian target of rapamycin (mTOR) pathways in PDAC cells. The results of the orthotopic animal model showed that pterostilbene combined with chloroquine significantly inhibited pancreatic cancer growth, delayed tumor quadrupling times, and inhibited autophagy and STAT3 in pancreatic tumors. In summary, the present study suggested the novel therapeutic strategy of pterostilbene combined with chloroquine against the growth of pancreatic ductal adenocarcinoma by inhibiting autophagy and downregulating the RAGE/STAT3 signaling pathways.

## 1. Introduction

Pancreatic ductal adenocarcinoma (PDAC) is the leading cause of cancer mortality worldwide. Only 10–15% of patients are considered eligible for surgery, and the remaining patients will receive first-line chemotherapy, in which gemcitabine (GEM) is the main chemotherapeutic drug used in first-line and in advanced pancreatic cancer patients [1]. More recently, GEM combined with nab-paclitaxel (GEM–NAB), as well as the combination of 5-fluorouracil, oxaliplatin, and irinotecan (FOLFIRINOX), have emerged as valid first-line therapies for advanced or metastatic pancreatic cancer (mPC) [2]. Pterillo et al. indicated that GEM–NAB was feasible and effective in patients previously treated with gemcitabine as an adjuvant treatment [3]. FOLFIRINOX is a form of combination chemotherapy for PDAC containing 5-FU, leucovorin, irinotecan, and oxaliplatin. This schedule is now also considered a standard treatment in first-line therapy, and improves overall and progression-free survival compared with GEM alone among patients with metastatic PDAC [4]. However, a systematic review indicated that the overall risk of death and progression were similar in patients receiving FOLFIRINOX as compared to GEM–NAB, despite a longer median overall survival being observed in the former [2]. The studies showed that the clinical course remains hampered by resistance to chemotherapy [5]. Therefore, development of novel therapeutic strategies for PDAC is still urgent.

The resistant characteristics of PDAC include Kirsten rat sarcoma virus (*KRAS*) gene mutations, which are present in 90% of PDAC tumors [6]. The oncogene *RAS* has been implicated in regulating a vast range of metabolic processes, including glucose uptake, glutamine metabolism, and increased autophagy [7]. Mutant *RAS* also activates the phosphoinositide 3-kinase (PI3K)/protein kinase B (AKT) pathways, which then activate mammalian target of rapamycin (mTOR), which is involved in the growth factor signaling and nutrient availability of PDAC [7]. Moreover, signal transducer and activator of transcription 3 (STAT3) overexpression, autophagy induction, and the interaction of immune cells and cancer stem cells that support the tumor microenvironment lead to GEM resistance, relapse, and death [1]. Despite the important role of the *RAS* and PI3K/AKT pathways in regulating the growth of PDAC, monotherapies targeting PI3K, AKT, and/or mTOR have been largely disappointing in *RAS*-mutant tumors [8]. In addition, previous reports indicate that ~5–10% of PDAC cases are hereditary in nature, and have DNA damage repair (DDR) mutations, including BRCA 1 and 2. These mutations could confer sensitivity to platinum-based therapies and determine eligibility for poly(ADP-ribose) polymerase inhibitors (PARPis), because the cells are unable to utilize the homologous recombination repair (HRP), leading to accumulation of double-stranded DNA breaks and, eventually, cell death [9]. Therefore, PDAC patients with breast cancer gene 1 or 2 (*BRCA* 1 or 2) mutations may derive greater benefit from platinum-based chemotherapy in first-line settings [10]. However, PARP inhibitor resistance has also been observed, thereby partially precluding their use in clinical applications. The major mechanism underlying this resistance could be the restoration of HRR [10]. Therefore, targeting multiple pathways—such as autophagy and survival signaling pathways—has the potential to lead to the development of new anticancer therapies for PDAC.

Among the above-mentioned alterations in signaling pathways in PDAC, autophagy has emerged as a key pathway promoting pancreatic carcinogenesis and resistance to chemotherapies [11]. Induction of autophagy is tightly regulated by various signaling pathways, such as inhibition of the PI3K/AKT and mTOR pathways, and activation of AMPK, autophagy proteins (Atgs)—including Atg8 (LC3), Atg5, Atg12, and Atg16L1—and lysosomal enzymes [12]. In the completion of autophagic flux, the LC3 and the cargo receptors—such as p62—are degraded and, therefore, can provide the assessment of autophagic flux [12]. Current findings further indicate that autophagy interacts with several survival signaling pathways in PDAC cells to promote growth and proliferation, and is associated with poor patient outcomes [13]. For instance, damage-associated molecular pattern (DAMP) molecule receptors such as high-mobility group box 1 (HMGB1) and the receptor for advanced glycation and products (RAGE) have been implicated in pancreatic carcinogenesis through the induction of autophagy [13,14]. Kang et al. further indicated that RAGE is an important inflammatory mediator that modulates crosstalk between autophagy and the pro-survival interleukin 6 (IL6)/STAT3 pathway in PDAC, in which autophagy contributes to cancer survival via activation of the STAT3 pathway and subsequent secretion of growth factors [14]. Multiple studies have also indicated that STAT3 is a central node in the development and progression of many human tumors, validating STAT3 as an anticancer target for PDAC [15]. Based on these findings, autophagy and its modulated signaling pathways—such as the RAGE/STAT3 pathways—are undoubtedly attractive targets for developing novel therapeutic strategies for PDAC.

Pterostilbene (trans-3,5-dimethoxy-4-hydroxystilbene; PT), which is classified as a stilbene compound, is a dimethyl ether analog of resveratrol [16]. PT is naturally produced by a variety of plants, including the Pterocarpus indicus, bilberries, blueberries, and almonds, as well as grape leaves and wines [16]. Research regarding PT has attracted more attention for its methoxy groups, which lead to better pharmacokinetic characteristics, including increased lipophilicity, better oral absorption, higher cellular uptake, and longer half-life (77.9 min) compared with resveratrol (10.2 min) [17]. PT has been studied for its excellent benefits in the prevention and treatment of various types of cancer, including breast, lung, bladder, and colon cancers [18,19]. The anticancer effects of PT include apoptosis, antioxidant and pro-oxidant effects, modulation of immune cells, cell cycle arrest, senescence, inhibition of angiogenesis and metastasis, and regulation of autophagy pathways [20,21]. In terms of the molecular mechanism of PT, previous reports have indicated that PT could inhibit tumor growth through the downregulation of Janus kinase/signal transducers 3 (JAK/STAT3), AKT/mTOR, and human telomerase reverse transcriptase (hTERT), and upregulation of the 5′-adenosine monophosphate activated protein kinase (AMPK), extracellular signal-regulated protein kinase 1/2 (ERK1/2), tumor protein P53 (p53), autophagy, nuclear factor erythroid 2-related factor 2 (Nrf2), and epigenetic regulated pathways [19,20,22,23,24,25]. More recently, clinical trials have been conducted to evaluate the safety of PT, and showed that PT is generally safe for use in humans up to 250 mg/day, and is also safe in treating or preventing cardiovascular disease [26]. In light of its safety profiles and beneficial chemopreventive effects in human beings, it is suggested that PT could potentially be used in cancer therapy for its modulating effects on multiple pathways in cancers.

PT has been shown to suppress proliferation and induce apoptosis in cancers, including PDAC cells, but the mechanisms involved are unclear [27,28,29]. A previous study indicated that the combination of chemotherapeutic agents with autophagy inhibitors such as chloroquine (CQ)—which prevents the fusion of autophagosomes and lysosomes and, therefore, inhibits the progress of autophagic flux—may induce damage to PDAC, leading to cancer cell death [30,31]. Therefore, the present study was undertaken to investigate the synergistic anticancer effects of PT combined with CQ, and the underlying mechanisms were evaluated using in vitro and in vivo orthotopic models. Our results showed for the first time that PT combined with CQ significantly inhibited autophagy, decreased cell viability, and sensitized the PDAC cells to PT-induced apoptosis via the downregulation of the RAGE/STAT3 and AKT/mTOR pathways. The combination of PT and CQ could act as a potential therapeutic strategy for PDAC.

## 2. Results

### 2.1. Pterostilbene Inhibits Growth in PDAC Cell Lines

To investigate the chemoresistant characteristics of pancreatic cancer cells against cancer therapeutic agents, the PDAC cell lines BxPC-3, PANC-1, MIA PaCa-2, and AsPC-1 were exposed to different concentrations of GEM (1, 5, 10, 25, 50, 100, and 150 μM) for 48 h. As shown in Figure 1A, BxPC-3 and AsPC-1 cells were more sensitive to GEM than PANC-1 and MIA PaCa-2 cells; however, cytotoxicity in all PDAC cells did not increase after escalating GEM treatment doses, suggesting that the chemoresistant characteristics occurred in PDAC cell lines (Figure 1A). The IC50 of GEM was 20, 110, 240, and 260 μM in BxPC-3, AsPC-1, PANC, and MIA PaCa-2 cells, respectively. Our previous studies indicated effective anticancer effects of PT in both sensitive and chemoresistant bladder cancer cells [20]. Accordingly, we evaluated the cytotoxic effects of PT (50, 75, 100, 125, and 150 μM) in PDAC cell lines. The results showed that all PDAC cell lines had similar sensitivity to PT treatment, in a dose-dependent manner (Figure 1B). The IC50 of PT in PDAC cell lines ranged from 110 to 130 μM, and the viability of BxPC-3 and MIA PaCa-2 cells after treatment with 150 μM PT was approximately 20% and 40%, respectively (Figure 1B). These results indicate that BxPC-3 cells are sensitive to GEM treatment, whereas MIA PaCa-2 cells are the most resistant (Figure 1A), indicating that the basal levels of survival signaling pathways may be different. Therefore, BxPC-3 and the most resistant cell line (MIA PaCa-2) were chosen for further studies to investigate whether inhibition of survival signaling could sensitize PDAC cells to PT treatment.

Next, we investigated the relative mechanisms regarding autophagy and survival components after PT treatment. As shown in Figure 1C, PT treatment for 48 h induced apoptosis (15–20%) and autophagy (20–40%), but no necrosis, in BxPC-3 cells (Figure 1C). Furthermore, PANC-1 (Figure 1D) and MIA PaCa-2 cells showed significantly increased autophagy, which was higher in MIA-PaCa-2 cells than others (approximately 80%), after PT treatment (Figure 1E). Time-course-dependent studies also confirmed that PT induced a significant increase in subG0/G1 (30%) in BxPC-3 cells compared to MIA PaCa-2 cells (15%), confirming that PT induced significant cell death in BxPC-3 cells, while MIA PaCa-2 cells were more resistant to PT treatment (Figure 1F,G).

### 2.2. The Combination of Pterostilbene and Chloroquine Exerts Enhanced Cytotoxicity against Pancreatic Cancer Cells

Because autophagy has been shown to contribute to chemoresistance in PDAC cells [14], we next investigated the autophagy levels in BxPC-3 and MIA PaCa-2 cells after PT treatment for 24 h. PT treatment caused the induction of autophagy in both PDAC cell types, as indicated by flow cytometry (Figure 2A,C) and immunofluorescence staining (Figure 2B,D). The results of Western blot analysis further showed a time-dependent induction of LC3-II and Beclin1 in both cell lines after PT treatment (Figure 2F,G). We also observed significantly reduced expression levels of Bcl-xl in PDAC cells in response to PT treatment.

In addition to the results of autophagy detection in Figure 2A–E, the results confirmed that the more resistant MIA PaCa-2 cells could more substantially induce autophagy after PT treatment, compared to BxPC-3 cells.

To overcome the effects of autophagy on cancer survival, we examined whether inhibition of autophagy could enhance the anticancer effects of PT in PDAC cells—especially in resistant MIA PaCa-2 cells. As shown in Figure 3A,D, 100 μM PT combined with the autophagy inhibitor CQ (5 or 10 μM) significantly inhibited cell viability in BxPC-3 (approximately 20%) and MIA PaCa-2 (approximately 40%) cells compared to PT or CQ treatment alone. Then, we detected the types of cell death in response to combined treatment. The results showed that treatment with CQ alone caused no statistically significant changes in apoptosis in either cell line, whereas PT combined with CQ significantly increased apoptosis, but not necrosis, in BxPC-3 cells (60%) and MIA PaCa-2 (30%) cells (Figure 3B,E). We also observed changes in the expression of proteins involved in apoptosis: decreased expression of Bcl-xl and increased expression of cleaved caspase-3 in both cell lines (Figure 3C,F). Furthermore, the accumulation of LC3 proteins (Figure 3C,F) and autophagosomes detected via TEM in MIA PaCa-2 cells confirmed that autophagic flux was inhibited by CQ (Figure 3G).

### 2.3. Pterostilbene Combined with Chloroquine Downregulates the RAGE/STAT3 Signaling Pathways and Increases Apoptosis

Previous studies have reported that STAT3 is constitutively activated in patients with pancreatic cancer, and is associated with therapeutic resistance [1]. It has also been reported that autophagy is required for the activation of the STAT3 pathway via RAGE signaling [13]. Therefore, we examined the protein expression of HMGB1, RAGE, and STAT3 in HPDE normal pancreatic cells and PDAC cell lines. The results showed that the expression of HMGB1, RAGE, p-STAT3 (Ser), and STAT3 was more prominent in MIA PaCa-2 cells compared to HPDE, BxPC-3, and other PDAC cell lines (Figure 4A). Treatment with either PT or CQ reduced the expression levels of HMGB1, RAGE, p-STAT3 (Ser), and STAT3 in MIA PaCa-2 cells, in a time-dependent manner (Figure 4B). In addition, combined treatment significantly potentiated the inhibitory effects on the expression of RAGE, p-STAT3 (Ser), and STAT3. Concurrently, apoptosis was increased in BxPC-3 and MIA PaCa-2 cells, as indicated by the increased expression levels of Bax and decreased Bcl-2 expression levels in response to PT combined with CQ (Figure 4C). The results further showed that caspase-8 and caspase-9 were activated in MIA PaCa-2 cells treated with PT or CQ alone, whereas caspase-9 activation was significantly increased in the combined treatment groups, confirming our previous finding that intrinsic apoptosis was activated after treatment with PT combined with CQ (Figure 4D). The results showed that, upon inhibition of autophagy, the RAGE/STAT3 signaling pathways were significantly inhibited, thereby increasing sensitivity to apoptosis by PT treatment.

To gain further insight into the anticancer mechanisms of combined treatment, we analyzed the expression of autophagy regulators—including the AKT/mTOR pathways—in BxPC-3 and MIA PaCa-2 cells in response to combined treatment. The activation of AKT and its downstream signaling pathways—including mTOR and p70—was more significantly inhibited in the combined treatment groups compared to either the PT or CQ treatment groups in BxPC-3 cells (Figure 4E). There was also a significant decrease in the activation of p38 and JNK, but not ERK1/2, in both cells—especially in BxPC-3 cells—after treatment with PT combined with CQ (Figure 4F). These results indicate that PT combined with CQ induced apoptosis by blocking autophagy, along with the inactivation of the RAGE/STAT3, AKT/mTOR, and JNK/p38 signaling pathways.

### 2.4. The Combination of PT and CQ Treatment Enhances Anticancer Effects in an Orthotopic Pancreatic Cancer Model

To investigate the utility of the combination treatment strategy in vivo, we established an orthotopic animal model in which SCID mice were injected into the pancreas with MIA PaCa-2-Luc cells containing stable luciferase activity. Orthotopically implanted SCID mice were treated with PT, CQ, or PT combined with CQ. Tumor growth was measured via IVIS imaging every week after tumor implantation. As shown in Figure 5A, no significant changes in body weight were observed, indicating that the treatments alone or in combination had a good safety profile. Bioluminescence imaging showed a gradual and time-dependent increase in tumor volume in the control groups, whereas the combined treatment groups showed significantly reduced tumor growth and tumor weight compared to the other groups (Figure 5B–E).

The tumor growth inhibition, tumor volume quadrupling times (TVQTs), and tumor growth delay times (TGDTs) are shown in Table 1. On the 18th day after the first treatment, tumor growth was inhibited by ~29.9%, ~19.7%, and ~39.4% in the PT, CQ, and combined treatment groups, respectively (Table 1). The TVQTs and TGDTs in the combined groups were ~33.5 and ~15.9 days, respectively, which were significantly different from the control and the individual treatment groups (*p* < 0.05) (Table 1).

Immunohistochemical staining demonstrated that the fractions of PCNA- and p-STAT3-positive cells in the combined treatment groups were significantly less than those in the PT or CQ monotherapy groups (Figure 6A). Moreover, the combined treatment groups exhibited increased accumulation of LC3 and increased expression levels of cleaved caspase-3 (Figure 6A). Western blot analysis confirmed that combined treatment significantly reduced the expression levels of RAGE, p-STAT3, STAT3, p-AKT, and PCNA in tumor tissues compared to the PT or CQ monotherapy groups (Figure 6B). Moreover, inhibition of autophagy was detected via the accumulation of p62 and LC3, and increased expression levels of cleaved caspase-3 along with the decreased expression of Bcl-xl and Bcl-2 in tumor tissues indicated that apoptosis was significantly induced after combined treatment (Figure 6C).

## 3. Discussion

Pancreatic cancer is one of the most lethal malignant tumors, with a high risk of metastasis and recurrence, and a poor 5-year survival rate (<5%) [11]. Conventional chemotherapy tends to fail due to poor responses and severe side effects. Therefore, a safer and more effective therapeutic strategy is needed for PDAC. In the present study, we demonstrated for the first time that PT combined with CQ targets autophagy to inhibit cancer cell growth and induce apoptosis in PDAC cells (Figure 6C). In response to the inhibition of autophagy by CQ, the AKT/mTOR pathway—which acts as the upstream mediator of autophagy—was inhibited. Moreover, accompanied by the inhibition of autophagy, RAGE/STAT3—which is an important pro-survival signaling pathway in PDAC cells—was significantly inhibited by the combined treatment (Figure 4 and Figure 6). Importantly, the anticancer effects of PT combined with CQ were translated into orthotopic animals, reaffirming the findings observed in PDAC cells (Figure 5 and Figure 6). Altogether, we suggest that PT combined with CQ could be an effective therapeutic strategy for PDAC, targeting autophagy and pro-survival signaling pathways such as the RAGE/STAT3 pathways, which are two of the most important survival pathways in PDAC.

Previous studies, including ours, have indicated that PT is a novel alternative medicine with diverse pharmacological benefits for the prevention and treatment of a vast range of human diseases. A previous study indicated that PT inhibits PDAC growth by downregulating glucocorticoid production, thus decreasing the Nrf2-dependent signaling/transcription and the antioxidant protection of PDAC [32]. Similar to our study, McCormack et al. also indicated that PT inhibits pro-survival signaling pathways, such as the STAT3 pathway in PDAC cells [29]. However, the anticancer effects and mechanisms of PT in PDAC cells remain largely unknown thus far. In this study, we observed that PT exerts anticancer effects against PDAC; however, pro-survival autophagy was induced in PDAC—especially in the more resistant MIA PaCa-2 cells—leading to less sensitivity to PT treatment (Figure 2). Numerous studies have indicated that the induction of autophagy in cancers cannot be ignored, due to its role in chemoresistance and cancer survival [33,34,35]. Therefore, in order to overcome chemoresistance in cancer cells, targeting autophagy via inhibitors could be a promising strategy for the treatment of PDAC. We then performed combined treatment with PT and the autophagy inhibitor CQ in PDAC cells, as well as in orthotopic animal models (Figure 4, Figure 5 and Figure 6). Our results indicate that PT combined with CQ restores sensitivity to apoptosis in resistant PDAC cells. Consistent with our findings, suppression of autophagy by CQ or its derivative hydroxychloroquine (HCQ) could sensitize PDAC cells to apoptosis induced by chemotherapeutic agents, such as GEM, sunitinib, or doxorubicin [36,37]. Mechanistic studies have shown that the inhibition of autophagy by CQ increases ROS, DNA damage, and mitochondrial defects, leading to decreased PDAC tumor progression in vitro and in vivo [6]. In addition, blockade of autophagy reduces pancreatic cancer stem cell activity, thereby potentiating the anticancer effects of GEM, with reduced phosphorylation of ERK and STAT3 [38]. A recent clinical trial indicated that HCQ combined with GEM is safe and well tolerated. The surrogate biomarker responses (CA 19-9) and surgical oncological outcomes were encouraging in patients with PDAC [39]. Another phase I study also indicated that the addition of CQ to GEM was well tolerated, and showed promising effects on the clinical response to the anti-pancreatic-cancer therapy [40]. However, the results of a clinical trial that used HCQ for previously treated metastases were disappointing, and the negligible therapeutic efficacy could be partially due to nonspecific autophagy inhibition effects of HCQ [41]. Altogether, the evidence indicates that autophagy inhibitors such as CQ could be used as an adjuvant therapy to chemotherapy in PDAC, offering more efficient tumor elimination and curative rates. In accordance with our findings, previous studies indicated that anticancer agents inhibited tumor growth, induced autophagy, and suppressed the JAK2/STAT3 pathway, while the autophagy inhibitor CQ enhanced this effect [42,43,44,45]; however, the mechanisms remain unclear, because the RAGE and STAT3 pathways are also regulated by various factors, including HMGB1, NF-κB, fibroblast-specific protein 1 (FSP1), SOCS3, CXCR3, etc. [46,47,48,49,50]. This could partially explain how PT combined with CQ might affect a complicated network in cells when regulating the RAGE and STAT3 pathways. Nevertheless, the mechanisms and the effects of combined CQ and anticancer agents for clinical usage should be further assessed.

Through the inhibition of autophagy, we observed that multiple survival pathways in PDAC were concomitantly inhibited by the combined treatment. Our results were similar to previous studies that have indicated a crosstalk between autophagy and other signaling pathways in PDAC cells. For instance, RAGE has been reported to promote PDAC survival in vitro and in vivo via DAMPs—such as HMGB1—sustaining autophagy and limiting apoptosis [51,52]. In addition, several diverse pathways—such as AKT/mTOR and STAT3—play essential roles in promoting the induction of autophagy [53]. Autophagy is also involved in the activation of three MAPKs, including p38, JNK, and ERK1/2, which have been shown to promote cell survival or apoptosis [54]. As mentioned above, all of these constitutively upregulated pro-survival pathways co-regulate autophagy. Interfering with these signaling pathways could inhibit autophagy, or vice versa, leading to apoptosis of PDAC cells [55]. Consistently, our results indicated that interfering with autophagy is associated with downregulation of the AKT/mTOR and RAGE/STAT3 pathways at 48 h, which could induce apoptosis of PDAC cells (Figure 4). Similarly, inhibition of autophagy in pancreatic cancer stem cells also reduced the phosphorylation of STAT3 [38]. Increasing evidence indicates that STAT3 inhibition in cancer cell lines could trigger growth arrest or apoptosis by PT [56,57]. Current findings also indicate that the induction of autophagy facilitates IL-6 secretion, suggesting a positive feedback loop for the IL-6/STAT3 pathway underlying the survival and drug resistance mechanisms, while inhibiting apoptosis [58,59]. In addition, inhibited autophagy activation by CQ, leading to the blockage of autophagic flux, thereby decreased the amount of IL-6, and inhibited STAT3 expression, which caused the shift from autophagy to apoptosis and increased the sensitivity of cells to cancer therapy [60]. Our results are similar to previous findings showing that PT combined with CQ significantly inhibited autophagy, decreased cell viability, and sensitized the cells to PT-induced apoptosis via downregulation of the RAGE/STAT3 pathways in PDAC cells. However, further investigations are still encouraged in order to delineate the cross-regulation mechanisms between autophagy, apoptosis, RAGE/STAT3, and MAPKs after treatment with PT combined with CQ in PDAC.

In addition to the in vitro studies, PT and CQ co-treatments inhibited autophagy and induced apoptosis in an orthotopic animal model (Figure 5 and Figure 6). The growth and the volume of orthotopic PDAC were significantly decreased in the combined treatment groups. We screened several pathways that have been shown to be important for PDAC cell survival for their potential roles in interacting with autophagy in tumors (Figure 6). Among the pathways targeted in our screening, the RAGE/STAT3 pathways stood out as having a potential pathway crosstalk with autophagy. To improve tumor sensitivity to PT, combined treatment with the autophagy inhibitor CQ could increase the sensitivity of PDAC cells to PT treatment. Our results indicated that the addictive effects of PT and CQ in combination are likely to be achieved, due to autophagy and RAGE/STAT3 inhibition leading to apoptosis. We concluded that PT is beneficial to health, with promising anticancer effects, and could be an ideal choice of alternative medicine for cancer therapy. It is of great importance to further evaluate the anticancer efficacy and the underlying mechanisms of PT combined with CQ in PDAC.

## 4. Materials and Methods

### 4.1. Chemicals

MTT (3-(4,5-dimethylthiazol-2-yl)-2,5-diphenyltetrazolium bromide), GEM, CQ, and PI (propidium iodide) were purchased from Sigma-Aldrich (St. Louis, MO, USA). Pterostilbene (96% purity) was a gift from Sabinsa Corporation (East Winsor, NJ, USA). Annexin V-FITC was purchased from BD Biosciences (San Jose, CA, USA).

### 4.2. Reagents

Primary antibodies against GAPDH, Bax, Bcl-2, Bcl-xl, p-AKT (ser), AKT, p-STAT3 (ser), STAT3, p-JNK, JNK, p-ERK, ERK, p-P38, P38, p-P70, P70, caspase-3, p-mTOR, mTOR, Beclin1, and PCNA were purchased from Cell Signaling Technology Inc. (Danvers, MA, USA). Anti-LC3 and anti-p62 antibodies were purchased from MBL International Corporation (Woburn, MA, USA). Antibodies against *RAS* and HMGB1 were purchased from Abcam (Cambridge, MA, USA). Horseradish peroxidase (HRP)-conjugated anti-mouse and anti-rabbit secondary antibodies were obtained from Jackson ImmunoResearch (West Grove, PA, USA).

### 4.3. Cell Culture

HPDE cells are normal pancreatic cells, which were provided by Professor Yan-Shen Shan (Institute of Clinical Medicine and Department of Surgery, College of Medicine, National Cheng Kung University, Tainan, Taiwan, and were cultured in keratinocyte SFM (Thermo Fisher Scientific Inc., Waltham, MA, USA). AsPC-1 (ATCC: CRL-1682) and BxPC-3 (ATCC: CRL-1687) cells were maintained in RPMI-1640 medium. PANC-1 (ATCC: CRL-1469) and MIA PaCa-2 (ATCC: CRL-1420) cells were maintained in DMEM. All media were supplemented with 100 U/mL of penicillin and 100 µg/mL of streptomycin (Gibco, Thermo Fisher Scientific Inc.), along with 10% heat-inactivated fetal calf serum (Thermo Fisher Scientific Inc.).

### 4.4. Cell Viability Assay

Cells were seeded in a 96-well plate at a density of 1 × 10^4^ cells/well, and incubated overnight. After removing the media, 100 μL of medium with GEM, PT, CQ, or PT combined with CQ was added at the indicated doses, followed by 48 h of incubation. After harvesting the cells at the indicated timepoints, viability was assayed via MTT assay.

### 4.5. Detection of SubG0/G1 and Apoptosis by Flow Cytometry

SubG0/G1 was detected by staining with PI. Apoptosis and necrosis were detected by staining with PI and Annexin V. Annexin-V-positive cells were apoptotic cells, while PI-positive cells were necrotic cells. All parameters were detected via flow cytometry and quantified using FlowJo 7.6.1. software (Tree Star Inc., Ashland, OR, USA).

### 4.6. Immunofluorescence Staining and Detection of Autophagy

BxPC-3 or MIA PaCa-2 cells were plated overnight in a six-well plate at a density of 1 × 10^5^ cells per well. After treatment with 100 μM PT for 72 h, the cells were fixed in 4% formaldehyde for 20 min, and then washed with phosphate-buffered saline (PBS). Autophagy was detected using the CYTO-ID^®^ Autophagy Detection Kit (Enzo Life Sciences, Inc., Farmingdale, NY, USA), which measures autophagosomes and autolysosomes in the cells, which were then analyzed using a fluorescence microscope or flow cytometry.

### 4.7. Western Blot Analysis

Pancreatic cancer cells or randomized frozen tumor samples from different treatment groups were homogenized, and lysates were subjected to gel electrophoresis and immunoblotting. Immunoreactive proteins were visualized with a chemiluminescent detection system (PerkinElmer Inc. MA, USA) and BioMax light film (Eastman Kodak Co., New Haven, CT, USA), according to the manufacturers’ instructions.

### 4.8. Transmission Electron Microscopy

Cells were fixed in a solution containing 2.5% glutaraldehyde and 2% paraformaldehyde (in 0.1 M cacodylate buffer, pH 7.3) for 1 h. After fixation, the samples were postfixed in 1% OsO_4_ in the same buffer for 30 min. Ultrathin sections were then observed under a transmission electron microscope (JEOL JEM-1200EX, Tokyo, Japan) at 100 kV.

### 4.9. Orthotopic Pancreatic Cancer Model

A luciferase-expressing MIA PaCa-2 cell line was established via transfection with a pGL-3 vector containing a luciferase-coding sequence (Promega, Madison, WI, USA), and selected by G418. The G418-resistant clones were referred to as MIA PaCa-2-Luc cells. Luciferase expression was confirmed by measuring the light emission after adding luciferin (150 μg/mL) (Promega Corporation, Madison, WI, USA), using an IVIS system (Xenogen Corp., Alameda, CA). Establishment of the orthotopic pancreatic cancer model was performed as described previously [61]. After having luciferase-expressing MIA PaCa-2 cells injected into their pancreases for 1 week, mice were treated with corn oil (control groups), PT (500 mg/kg, dissolved in corn oil), CQ (10 mg/kg), or PT combined with CQ by oral gavage for 28 days (*n* = 5 in each group). After treatment, mice were euthanized, and the tumors were collected, weighed, and preserved in 10% formalin for further analysis.

### 4.10. H&E Staining and Immunohistochemistry Staining

Serial tissue sections (4 μm) of pancreatic tumors were sliced from paraffin-embedded tissues and stained with hematoxylin and eosin (H&E). Immunohistochemistry staining assays were performed using a STARRTREK Universal HRP Detection Kit (Biocare Medical, Concord, CA, USA) according to the manufacturer’s protocol, and the slides were counterstained with hematoxylin. The percentage of positive-stained area was calculated using FlowJo software, and then compared with the whole area.

### 4.11. Statistical Analysis

The results are presented as the mean ± standard error of the mean (SEM). Experimental data were analyzed using Student’s t-test. Differences were considered to be statistically significant when the *p*-value was less than 0.05.

## Figures and Tables

**Figure 1 molecules-26-06741-f001:**
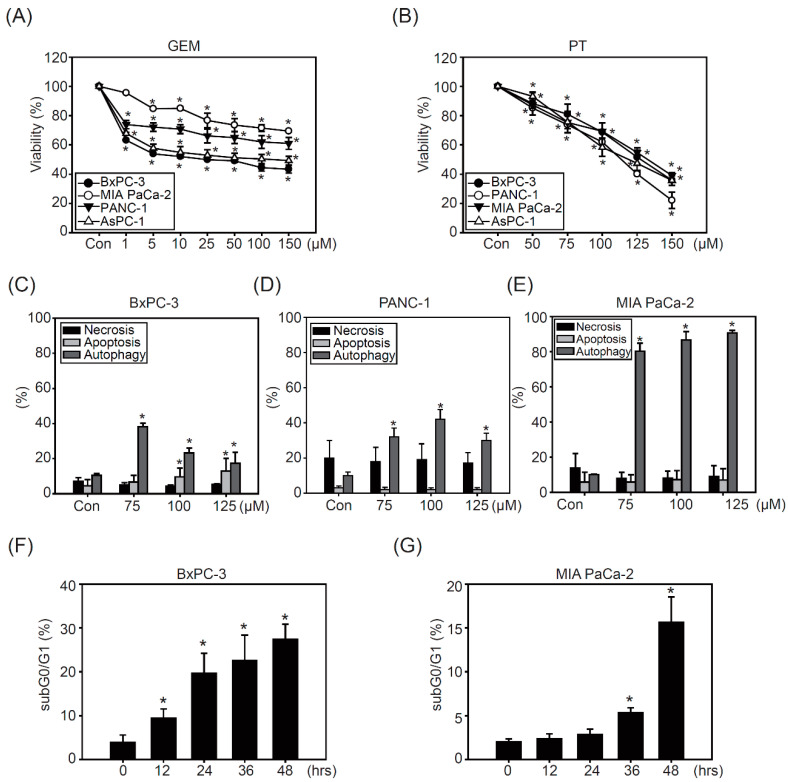
The cytotoxic effects of pterostilbene (PT) in pancreatic cancer cells: (**A**) Gemcitabine (GME) (0, 1, 5, 10, 25, 50, 100, and 150 μM) and (**B**) pterostilbene (PT) (0, 50, 75, 100, 125, and 150 μM) treatment on BxPC-3, PANC-1, AsPC-1, and MIA PaCa-2 cells for 48 h. Cell viability of four cell lines was analyzed via MTT assay. The data are presented as the mean ± SEM, *n* = 3. * *p* < 0.05 compared to the control (DMSO) groups. Necrosis, apoptosis, and autophagy analysis were performed via flow cytometry in (**C**) BxPC-3, (**D**) PANC-1, and (**E**) MIA PaCa-2 cells treated with PT in a concentration-dependent manner (0, 75, 100, and 125 μM) for 48 h. Percentage of subG0/G1 was analyzed in (**F**) BxPc-3 and (**G**) MIA PaCa-2 cells treated with PT (100 μM) for 0, 12, 24, 36, and 48 h, and then detected with propidium iodide (PI) staining and analyzed via flow cytometry. The data are presented as the mean ± SEM; *n* = 3. * *p* < 0.05 compared to the control (Con, DMSO) groups.

**Figure 2 molecules-26-06741-f002:**
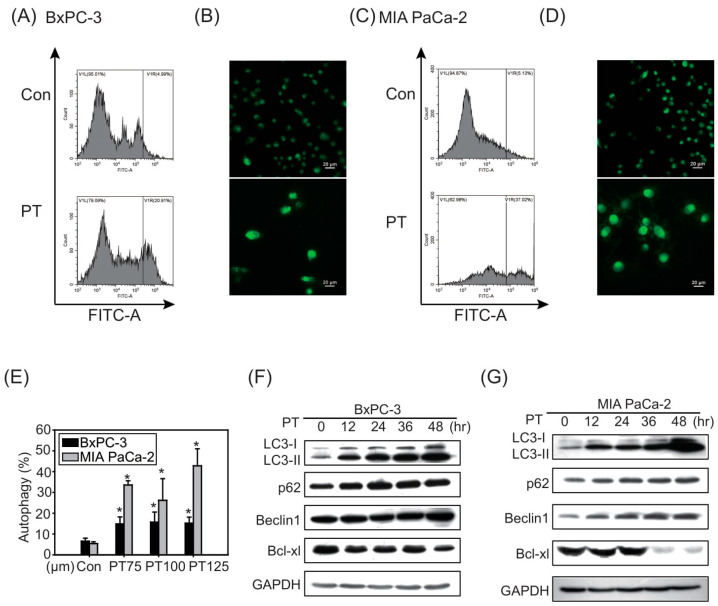
Autophagy was induced in response to PT treatment. The development of AVOs (acidic vesicular organelles) in (**A**) BxPC-3 and (**C**) MIA PaCa-2 pancreatic cancer cells after PT treatment for 24 h was analyzed via flow cytometry and (**E**) histogram indicate the percentage of autophagy positive cells via flow cytometry; * *p* < 0.05 compared to the control group. (**B**,**D**) Detection of autophagy in both cell lines via fluorescence microscopy at 400× magnification (scale bar = 50 μm). Western blot analysis of LC3-I, LC3-II, p62, Beclin 1, and Bcl-xl was conducted in (**F**) BxPC-3 and (**G**) MIA PaCa-2 cells treated with PT (100 μM) for 48 h. The membrane was probed with anti-GAPDH to confirm equal loading of proteins. Immunoblots are representative of at least three independent experiments.

**Figure 3 molecules-26-06741-f003:**
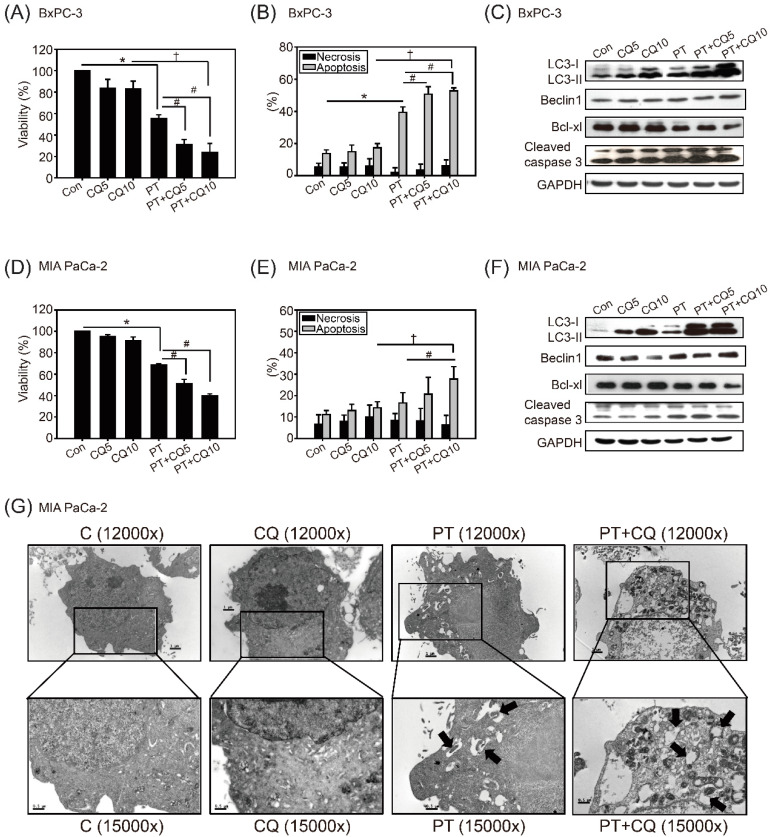
Synergistic cytotoxic effects of PT combined with the autophagy inhibitor chloroquine (CQ). Dose-dependent cytotoxic effects of CQ (5, and 10 μM) and PT (100 μM) treatment alone or in combination (PT + CQ) in (**A**) BxPC-3 cells and (**D**) MIA PaCa-2 cells for 48 h, analyzed via MTT assay. The data are presented as the means ± SEM of three independent experiments. * *p* < 0.05 compared to the control group; # *p* < 0.5 compared to the PT treatment alone groups; † *p* < 0.05 compared to the CQ 10 μM groups. Necrosis and apoptosis were analyzed via flow cytometry with Annexin V/PI in (**B**) BxPC-3 and (**E**) MIA PaCa-2 cells that were treated with CQ and PT—alone or in combination—for 48 h. The data are presented as the mean ± SEM; *n* = 3. * *p* < 0.05 compared to the control group; # *p* < 0.5 compared to the PT treatment alone groups; † *p* < 0.05 compared to the CQ 10 μM groups. Autophagy accumulation in (**C**) BxPC-3 and (**F**) MIA PaCa-2 cells after treatment with CQ alone or PT combined with CQ was analyzed by the expression of LC3-I and LC3-II, via Western blot analysis. Apoptosis-inducing effects were analyzed by the expression of Bcl-xl and cleaved caspase-3. The membrane was probed with anti-GAPDH to confirm equal loading of proteins. Immunoblots are representative of at least three independent experiments. (**G**) Electron micrograph of MIA PaCa-2 cells treated with 10 μM CQ and 100 μM PT—alone or in combination—for 48 h (C, CQ, PT, and PT + CQ, 12,000× magnification; and C/2, CQ/2, PT/2, and PT + CQ/2, 15,000× magnification). The arrows indicate autophagic vacuoles and autolysosomes. Scale bar = 1 μm.

**Figure 4 molecules-26-06741-f004:**
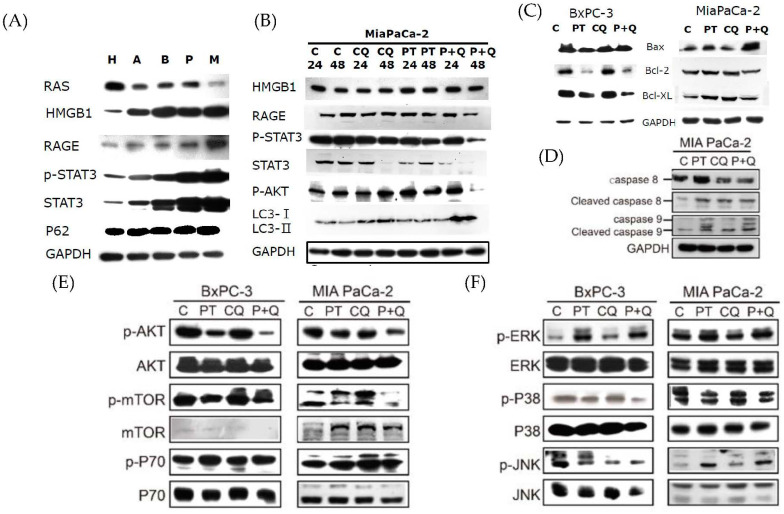
PT combined with CQ downregulates autophagy and the RAGE/STAT3 signaling pathways, leading to apoptosis: (**A**) The RAS, HMGB1, RAGE, p-STAT3, STAT3, and p62 protein expression in HPDE normal human pancreatic cells (H) and in AsPC-1 (A), BxPC-3 (B), PANC-1 (P), and MIA PaCa-2 (M) pancreatic cancer cells. (**B**) The effects of CQ and PT treatment—alone or in combination—for 24 and 48 h on the expression of HMGB1, RAGE, p-STAT3, STAT3, and LC3-I/II in MIA PaCa-2 cells. (**C**,**D**) BxPC-3 cells and MIA PaCa-2 cells were treated with CQ and PT—alone or in combination—for 48 h; the apoptosis-related proteins Bax, Bcl-2, and Bcl-xl were detected in both cell lines, and caspase-8, cleaved caspase-8, caspase-9, and cleaved caspase-9 were detected in MIA PaCa-2 cells. (**E**,**F**) The signaling pathways—including AKT, p-AKT, mTOR, p-mTOR, p70, p-p70, ERK, p-ERK, P38, p-P38, JNK, and p-JNK pathways—were examined in both cells via Western blotting. The membrane was probed with anti-GAPDH to confirm equal loading of proteins. Immunoblots are representative of at least three independent experiments.

**Figure 5 molecules-26-06741-f005:**
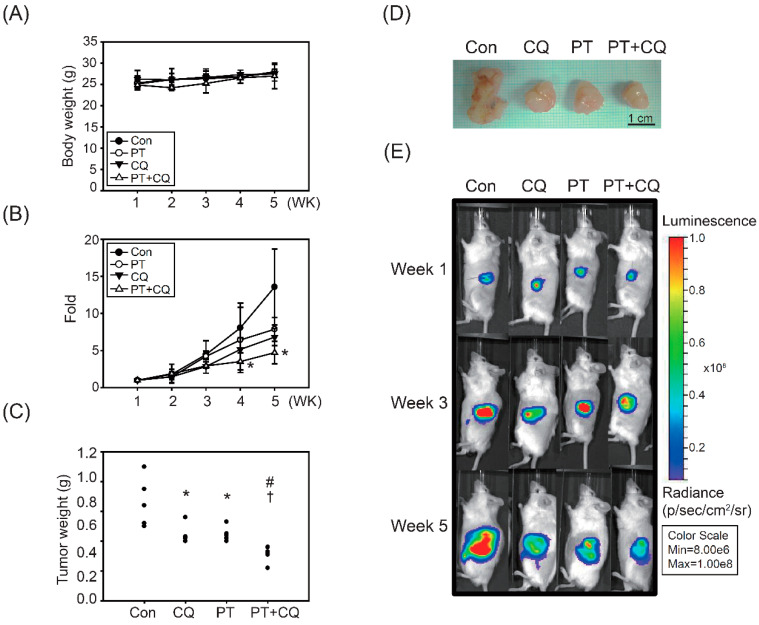
Combination of PT and CQ treatment enhances anticancer effects in the orthotopic pancreatic cancer model. MIA PaCa-2-Luc cells with luciferase activity were injected into the pancreas and administered with CQ (10 mg/kg) or PT (500 mg/kg)—alone or in combination—for 28 days. (**A**) Body weight in SCID mice was measured once per week. (**B**) Quantitative analysis of bioluminescence photon counts, as a measure of tumor growth throughout the entire experiment. * *p* < 0.05, compared to the control groups. (**C**,**D**) Measurement of tumor weight of orthotopic SCID mice after euthanasia. Data are presented as the mean ± SD (*n* = 5 mice/group). * *p* < 0.05 compared to the control groups; # *p* < 0.05 compared to the CQ treatment groups; † *p* < 0.05 compared to the PT treatment groups. Scale bar = 1 cm. (**E**) Quantification of tumor burden by measuring luciferase in the pancreas. Direct observation of SCID mice with tumors via in vivo imaging of luciferase activity in the pancreas.

**Figure 6 molecules-26-06741-f006:**
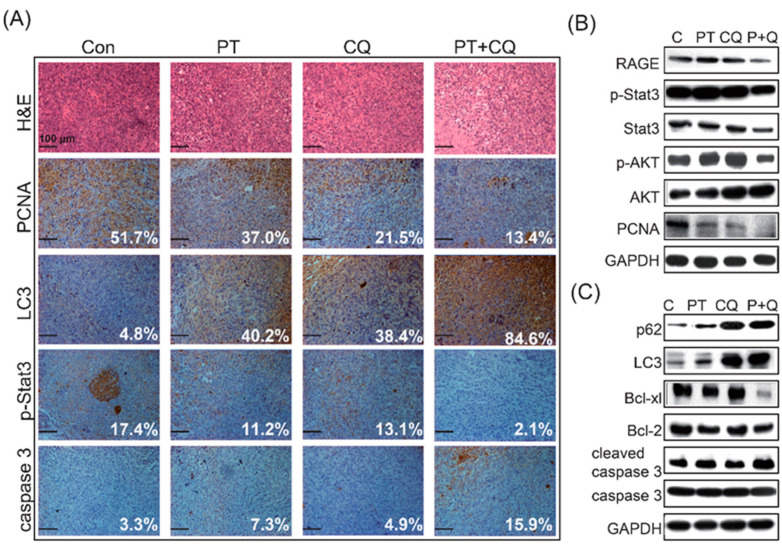
The combination of PT and CQ induces the expression of apoptosis-related proteins, and downregulates the expression of autophagy and RAGE/STAT3 signaling pathways in tumor tissues. (**A**) Pancreatic tumors were examined via H&E staining and immunohistochemistry to analyze the expression levels of PCNA, LC3, p-STAT3, and caspase-3, using FlowJo software. Scale bar = 100 μm. (**B**) PT and CQ treatment—alone or in combination—decreased the expression levels of RAGE, p-STAT3, STAT3, p-AKT, AKT, and PCNA in tumor tissues, while inhibition of autophagy was determined by the accumulation of p62 and LC3 (**C**). (**C**) Apoptosis in tumors treated with PT and CQ—alone or in combination—showed the expression of p62, LC3, Bcl-xl, Bcl-2, cleaved caspase-3, and caspase-3 via Western blotting. The membrane was probed with anti-GAPDH to confirm equal loading of proteins. Immunoblots are representative of at least three independent experiments.

**Table 1 molecules-26-06741-t001:** Comparison of tumor growth inhibition, tumor volume quadrupling times, and tumor growth delay times in orthotopic SCID mice implanted with MIA PaCa-2-Luc cancer cells.

Groups	Number of Mice	Inhibition (%)	TVQTs	TGDTs	*p*-Value (*t*-Test)		
					Control	CQ	PT
Control	5	--	17.6 ± 2.3	--			
CQ	5	29.9 ± 12.8	24.1 ± 4.2	6.5 ± 4.2	* <0.05		
PT	5	19.7 ± 13.4	21.3 ± 3.4	3.6 ± 3.4	0.08	0.27	
CQ + PT	5	39.4 ± 17.2	33.5 ± 12.0	15.9 ± 12.0	* <0.05	0.14	# <0.05

Data are presented as mean ± SEM; *n* = 5. * *p* < 0.05 compared to the control group on day 18; # *p* < 0.05, significance estimated by Student’s t-test in comparison with tumor growth delay times of the control group. TVQTs: tumor quadrupling times (days); TGDTs: tumor growth delay times (days).

## Data Availability

Not applicable.

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
