# Peer review of "Chloroquine Potentiates the Anticancer Effect of Pterostilbene on Pancreatic Cancer by Inhibiting Autophagy and Downregulating the RAGE/STAT3 Pathway"

_molecules, 2021, doi:10.3390/molecules26216741_

Round 1

Reviewer 1 Report

The submitted review titled " Combination of Pterostilbene and Chloroquine Enhances Anti-Cancer Effects in Pancreatic Cancer through Downregulation of Autophagy and RAGE/Stat3 Pathway" summarizes the importance of treatment with Pterostilbene in combination with chloroquine as a novel therapeutic strategy for pancreatic ductal adenocarcinoma.

Major concerns:

  1. The abbreviations shouldn’t be used in the abstract.
  2. What markers were used for the detection of apoptosis, autophagy and necrosis by flow cytometry? Should be mentioned in the main text.
  3. Did the authors determine IC50 for each used substances?
  4. Why the authors didn’t perform a cytotoxicity test for Chloroquine with all cell lines?
  5. Figure 1 – the proliferation and viability were performed at which time; 24, 48 or 72 hours? It is confusing and doesn’t correspond with the methodology.
  6. Why did the authors choose concentrations 75, 100 and 125 µM? Based on what?
  7. There is not a clear explanation why the MIA PaCa-Luc cell line was used instead of any of the other tested cell lines.
  8. Did the authors perform an MTD (maximum tolerated dose) for the tested substances in mice? If not, why did the authors choose a concentration of 10 mg/kg and 500 mg/kg? Based on which results?
  9. Why did the authors treat the mice only for 28 days?
  10. How did the authors score (% in figures) the tissue samples from the mice?
  11. Table 1 – P value is < 0.05 or ≤ 0.05?
  12. The text describing the figures should be more concise. Almost all of the text should be placed in the main text or in the material and methods section.
  13. The mice survival chart after treatment is missing.
  14. The H&E figures should be larger.
  15. The material and methods section should be more concrete. Chemicals and reagents should be placed in separate sub-sections.
  16. When were the tumors collected? How long after treatment?

Reviewer 2 Report

I have gone through the manuscript. Data is indeed impressive. However, It needs modifications. Authors claimed that Pterostilbenes induced activation of caspase-3. However, caspase-3 is the end product. It will be better if authors can provide experimental evidence about extrinsic or intrinsic apoptosis. Evaluation of caspase-8 and caspase-9 will be helpful. 

How strong is the chance of the anti-metastatic role of pterostilbenes? How about the evaluation of the anti-metastatic role of pterostilbene in the inhibition of the metastatic spread of tumor-bearing mice. 

Reviewer 3 Report

Reviewer comments and suggestions

The manuscript “Combination of Pterostilbene and Chloroquine Enhances Anti-Cancer Effects in Pancreatic Cancer through Downregulation of Autophagy and RAGE/Stat3 Pathway” is within the scope of Molecules. This article is well written. However, in my opinion, manuscript suffers from some serious weaknesses that should be corrected before publication. Below please find my comments and suggestions.

Title: RAGE/Stat3 Pathway, please check it has written correctly.

Abstract:

  1. Line 35, 37 and 39, please confirm and check Stat3 has written correctly.
  2. Line 35 AKT/mTOR, if first time please introduce it

Introduction:

  1. Line 61 KRAS mutations, if first time please introduce it.
  2. Line 64-65, Mutant RAS also activates PI3K/AKT pathways which then activates mTOR please abbreviate it if first time used.
  3. In addition, previous 70 reports indicated that about 5–10% PDAC cases are hereditary in nature and have DNA damage repair (DDR) mutations including BRCA 1 and 2, if first time please introduce it.
  4. Line 86, PI3K/Akt, some places you write PI3K/AKT, please write it consistently.
  5. line 118-119, ERK1/2, p53, autophagy, Nrf2, and epigenetic regulated pathways, if first time please introduce it mean abbreviation.

Results:

  1. Why authors choose 48 hrs time period? Also I am curious to know its time dependent or concentration dependent experiment.
  2. Line 287, *P<0.05, please check it has written correctly.
  3. Line 225,334,339,390,413,421,432, RAGE/Stat3, some places you write RAGE/STAT3, please make it consistent. throughout of the manuscript.

Discussion

  1. Line 410, IL-6 secretion, if first time used please abbreviate it.

Materials and methods

  1. Line 478, washed with PBS, if first time used abbreviate it.

References

  1. Ref. 4, 12, 20, 25, 36, 43, 46, 57, 60, use journal name abbreviation not full form.
  2. Also check all the references format accordingly journal guideline.

Drubay, V.; Skrypek, N.; Cordiez, L.; Vasseur, R.; Schulz, C.; Boukrout, N.; Duchene, B.; Coppin, L.; 526 Van Seuningen, I.; Jonckheere, N., TGF-betaRII knock-down in pancreatic cancer cells promotes tumor 527 growth and gemcitabine resistance. importance of STAT3 phosphorylation on S727. Cancers 2018, 10, 528 (8), doi:10.3390/cancers10080254.

Caparello, C.; Meijer, L. L.; Garajova, I.; Falcone, A.; Le Large, T. Y.; Funel, N.; Kazemier, G.; Peters, G. 538 J.; Vasile, E.; Giovannetti, E., FOLFIRINOX and translational studies: Towards personalized therapy in 539 pancreatic cancer. World journal of gastroenterology 2016, 22, (31), 6987-7005.

Reviewer 4 Report

Title: Using the term ‘downregulation’ for a process (autophagy) and pathway (RAGE/Stat3) in the same sentence is a bit awkward. Authors are suggested to modify it. For example, ‘……..downregulation of autophagy targeting the RAGE/Stat3 pathway’. Phrases like ‘enhances anticancer effects in pancreatic cancer’ are not very smart as well.  

Abstract: The first sentence is incorrect and unclear. Half of the abstract was written to write the aim of study. Please include some results to improve the abstract.

Introduction:

Line 61-62: “The resistant characteristics of PDAC include KRAS mutations, which are presents in 90% of PDAC tumors [6].” Please correct the sentence.

Results:

Line 135-136: “The combination ….. could be act ……. for PDAC”. Please correct the sentence.

Line 143: “…..did not increased…..”. Please correct the sentence.

Line 149-151: Should it be BXPC-3? Or PANC-1? Please check this statement.

Line 157-159: “Furthermore, MIA PaCa-2 cells showed slightly increased in apoptosis but significantly increased in autophagy (approximately 80%) after PT treatment (Fig. 1D)”. Please correct the sentence.

Figure 2 (F, G): MIA PaCa-2 showed higher percentages of autophagy than BxPC-3 in 1 (C, D) or 2E. Should not the expression of Belin-1 be higher in 2G than 2F? In case of apoptosis, expression of caspase 3 is higher for BxPC-3 than MIA PaCa-2 (3C, 3F) which agrees with the findings shown in 3B, 3E. Can the authors explain it?

Figure 3 (B, E): What is the reason behind the increase of percentage of apoptosis when you apply Chloroquine, an autophagy inhibitor with PT?

Discussion:

Line 412-415: “Thus, inhibited autophagy activation by using CQ leading to blockage of autophagic flux thereby decreased the amount of IL-6 and inhibited Stat3 expression, which caused the shift from autophagy to apoptosis and increased the sensitivity of cells to cancer therapy [60]”. Authors did not perform experiments on IL-6 expression in this study though there are supporting articles (reference no. 58 and 59). So, they should not make such concluding remarks and are suggested to modify this particular sentence.

Line 427-428: “Among the pathways targeted in our screening, the RAGE/Stat3 pathways stood out as a potentially pathways crosstalk to autophagy”. Please correct the sentence. There are some more grammatically incorrect sentences. Please check the manuscript carefully.

Information on PT from the second last paragraph of Introduction (Line 104-124) was repeated in the second paragraph of Discussion. Please check it and modify. Authors should also brush-up the discussion to avoid redundant mentioning of positive effects of the combined use of PT and CQ. The last paragraph should be shortened excluding the statement containing KRAS mutation (Line 436-443). 

Round 2

Reviewer 1 Report

The authors answered all questions with valid points.

Reviewer 2 Report

Manuscript looks more comprehensive now...